



# Bayesian inverse modeling of the atmospheric transport and emissions of a controlled tracer release from a nuclear power plant

Donald D. Lucas, Matthew D. Simpson, Philip Cameron-Smith, and Ronald L. Baskett

Lawrence Livermore National Laboratory, Livermore, CA, 94550, USA.

*Correspondence to:* D. D. Lucas (ddlucas@llnl.gov, ddlucas@alum.mit.edu)

**Abstract.** Probability distribution functions (PDFs) of model inputs that affect the transport and dispersion of a trace gas released from a coastal California nuclear power plant are quantified using ensemble simulations, machine learning algorithms, and Bayesian inversion. The PDFs are constrained by observations of tracer concentrations and account for uncertainty in meteorology, transport, diffusion, and emissions. Meteorological uncertainty is calculated using an ensemble of simulations of the Weather Research and Forecasting (WRF) model that samples five categories of model inputs (initialization time, boundary layer physics, land surface model, nudging options, and reanalysis data). The WRF output is used to drive tens of thousands of FLEXPART dispersion simulations that sample a uniform distribution of six emissions inputs. Machine learning algorithms are trained on the ensemble data, and used to quantify the sources of ensemble variability and to infer, via inverse modeling, the values of the 11 model inputs most consistent with tracer measurements. We find a substantial ensemble spread in tracer concentrations (factors of 10 to $10^3$), most of which is due to changing emissions inputs (about 80%), though the cumulative effects of meteorological variations are not negligible. The performance of the inverse method is verified using synthetic observations generated from arbitrarily selected simulations. When applied to measurements from a controlled tracer release experiment, the most likely inversion results are within about 200 meters of the known release location, 5 and 50 minutes of the release start and duration times, respectively, and 22% of the release amount. The inversion also estimates probabilities of different combinations of WRF inputs of matching the tracer observations.

## 1 Introduction

Although the probability of a nuclear power plant accident is low, the risks associated with accidental releases of radioactive materials from nuclear power plants are expected to remain elevated worldwide through the coming decades (Christoudias et al., 2014). In the unlikely event of an accident, government agencies and plant owners must take actions to protect people and the environment from exposure to radioactive contamination. Because the atmosphere can spread the contaminants beyond the boundaries of a power plant within a matter of minutes to hours, reliable and timely protective action recommendations based on numerical modeling of actual releases are essential.

A variety of atmospheric models have been developed for simulating the transport and dispersion of releases from nuclear power plants, starting from the accidents at Three Mile Island in 1979 and Chernobyl in 1986 (e.g., Wahlen et al., 1980; Albergel et al., 1988; Gudiksen et al., 1989). These models range from simple straight-line Gaussian plumes that are applicable



at short ranges when the turbulence in the atmosphere is stationary and homogeneous (Seinfeld and Pandis, 2006), to more sophisticated models based on Lagrangian particles and/or Eulerian transport when the atmospheric flow is unsteady and occurs in areas with complex terrain (e.g., Pöllänen et al., 1997; Lauritzen and Mikkelsen, 1999; Nasstrom et al., 2007; Suh et al., 2009; Brioude et al., 2013b).

Atmospheric models used for nuclear power plant applications also use emissions modules to estimate the release rates of radionuclides based on specific reactor conditions (Athey et al., 1999). While these modules are useful for providing approximate ranges of emissions and their associated consequences, the detailed reactor conditions during an accident may not be well known and can contribute significant uncertainty to transport and dispersion predictions. The amount of radionuclides released to the atmosphere during the Fukushima Daiichi accident in 2011, for example, still remains highly uncertain because
electrical power was lost and the reactors were monitored only indirectly (e.g., Chino et al., 2011; Terada et al., 2012; Stohl et al., 2012; Katata et al., 2015).

Inverse modeling can provide a safe way to infer information about radioactive emissions from nuclear power plants, and can also help estimate uncertainty in the meteorological fields used to transport the radioactive materials. Emissions and winds are constrained in an inverse method by minimizing differences between dispersion model predictions and observations of
materials transported and deposited downwind from the source location (e.g., Davoine and Bocquet, 2007; Zheng and Chen, 2011). Building upon our previous work using Bayesian inverse modeling to estimate regional-scale greenhouse gas emissions (Lucas et al., 2015) and meteorological uncertainty in an urban-scale dispersion experiment (Lucas et al., 2016), we developed an ensemble-based inverse modeling system for analyzing nuclear power plant dispersion events. A diagram of the system is presented in Fig. (1) and summarized below.

Starting from the left hand side of the diagram in Fig. (1), an ensemble of plausible meteorological fields is generated using the Weather Research & Forecasting (WRF) model (Skamarock et al., 2008; Skamarock and Klemp, 2007). Ensemble members in WRF are created using different reanalysis datasets, physics packages, and configuration options, which are represented as categorical random variables. Sect. 2.1 provides further details of the WRF ensemble setup and design. The output of the WRF ensemble is then used to drive an ensemble of FLEXPART dispersion plumes (Brioude et al., 2013a), which also
considers variations in the location, timing, and magnitude of emissions using continuous random variables. Further details of the FLEXPART calculations are given in Sect. 2.2.

The WRF-FLEXPART ensemble provides a set of plume predictions that are compared with field measurements (right hand side of Fig. 1). The differences between the simulations and field data are minimized through Monte Carlo sampling loops that jointly vary the inputs to WRF and emissions in FLEXPART (red dashed lines in the diagram). Because a single
iteration through the sampling loops is computationally expensive and millions of iterations may be needed, we use machine learning to accelerate the optimization. Moreover, by sampling the WRF and FLEXPART inputs with a probability distribution function, Bayesian analysis is used to estimate uncertainty in the model inputs and outputs. Details of the machine learning and inversions methods are provided in Sects. 3.1 and 3.2.

Measurements from a tracer release experiment conducted (Thuillier, 1992) in September 1986 at the Diablo Canyon nuclear
power plant are used to test and verify the atmospheric models and inversion system in Fig. (1). Diablo Canyon is located



along the rugged coast of California (see Fig. 2), so the study provides a critical test of simulating transport and dispersion in complex terrain. The study also provides an important verification test of the inversion algorithm, because the tracer emissions are assumed to be unknown and inferred in the inversion. Information about the Diablo Canyon study is given in Sect. 4.

To our knowledge, this work represents the first joint inversion capability applied to FLEXPART dispersion simulations that
provides probability distribution functions of categorical inputs in WRF and continuous inputs in FLEXPART, and has been verified with tracer release data. This capability can be useful for other applications beyond releases from nuclear power plants, including compliance monitoring of the nuclear test ban treaty (Issartel and Baverel, 2003) and inverse modeling of emissions from large-scale industrial accidents and volcanic plumes (e.g., Heng et al., 2016).

## 2   Ensemble atmospheric modeling

### 2.1   Weather Research & Forecasting Model

The non-hydrostatic, fully compressible Weather Research and Forecasting (WRF) atmospheric model (Skamarock et al., 2008; Skamarock and Klemp, 2007) is used to generate meteorological fields for the atmospheric transport and diffusion simulations. Version 3.6.0 of the advance research WRF (ARW) core is used for the simulations presented in this paper. WRF was developed through collaboration among government, research, and academic organizations to facilitate the transfer of
state-of-the-science atmospheric research findings to an operational modeling capability. The National Center for Atmospheric Research currently maintains the open source WRF model code, which is publically available for user download. WRF is widely used by numerous groups for both atmospheric research and real-time operational commercial applications, such as renewable generation and utility grid demand forecasting.

### 2.1.1   WRF Domain

High resolution winds are needed to simulate the dispersion for the Diablo Canyon tracer release test problem (see Fig. 2 and Sect. 4). Through efficient numerical model nesting and the parallelization of source code for high performance computers, WRF can be used to simulate a large range in scales of motion from thousands of kilometers down to tens of meters (Lundquist et al., 2010).

Using this nesting capability, five WRF model domains are used to downscale and generate high-resolution meteorological
fields over the Diablo Canyon region. The WRF domain configuration and geographic coverage are shown on the left hand side of Fig. 3. A large portion of the Western United States is covered by the outermost model domain (labeled D1) at 24.3 km horizontal grid spacing. The large outer model domain was required for downscaling purposes given the coarse resolution of some of the reanalysis data sets used to initialize WRF for this study.

By downscaling, a horizontal grid spacing of 300 m is achieved in the innermost WRF model domain (D5, right hand side
of Fig. 3). The fine-scale grid spacing of D5 is necessary to generate representative meteorological conditions since the Diablo Canyon nuclear power plant is located on the coast near narrow valleys and other topographic features that result in complex





terrain induced flow. High-resolution terrain and land use fields were generated for D5 by downloading National Elevation Dataset and National Land Cover Database at 1-arc second (approx. 30 m) data from the United States Geological Survey Multi-Resolution Land Characteristics data server (Homer et al., 2015).

### 2.1.2 WRF Ensemble

Several features also make WRF ideal for creating an ensemble of plausible atmospheric conditions for uncertainty assessments. Ensemble modeling approaches have been shown to be effective at quantifying physically plausible states of the atmosphere in a probabilistic manner (e.g., Mullen and Baumhefner, 1994; Stensrud et al., 2000; Berner et al., 2011). The WRF modeling system contains numerous physics schemes that parameterize subgrid-scale processes, such as the surface energy exchange, cloud microphysics, and turbulent mixing in the planetary boundary layer. By running WRF with multiple

combinations of physics options, an ensemble is generated that captures meteorological uncertainty due to subgrid-scale parameterization error. In addition, WRF can be initialized and run using a variety of publically available reanalysis data sets to quantify meteorological uncertainty resulting from errors in initial / lateral boundary conditions. Meteorological data from gridded analysis data sets and observations can also be integrated into WRF simulations to improve model accuracy by using four dimensional data assimilation (FDDA) options for analysis (Stauffer and Seaman, 1994) and observational (Liu et al.,

2005, 2009) nudging. The goal of the analysis FDDA option is to nudge large-scale motion towards an observed state using relaxation terms, while the observational nudging impacts the prediction of local-scale atmospheric phenomenon.

Table 1 summarizes the five major variables that were selected for the WRF ensemble. For the purposes of the Monte Carlo sampling and analysis (Fig. 1), the WRF variables are treated as categorical random variables. By taking all of the combinations among the five variables in the table, we constructed a WRF ensemble containing 162 members. The major categories in the

table include model initialization time, source of input reanalysis data, FDDA nudging weighting factors, planetary boundary layer (PBL) physics, and land surface model (LSM) physics. The ensemble categories and their variations were selected based on previous WRF user experience and a literature review of sources of uncertainty that are likely to impact meteorological fields important to the specific tracer release experiment described in Sect. 4. Variations in PBL and LSM physics, for example, have been shown to affect near surface stability and wind fields (Lee et al., 2012), which can have a large impact on plume

transport modeling. Other categories related to changes in microphysics and cumulus physics are not considered in this report because precipitation and cloud cover were not present during the specific tracer release experiment. Future studies can easily incorporate these factors, and others, by including additional categories for Monte Carlo sampling.

Several variations are included in the weather ensemble to account for uncertainty related to model initialization and meteorological reanalysis inputs. WRF simulations were started at either 15 hours or 9 hours before the beginning of the tracer

release (i.e., at 00:00 UTC or 06:00 UTC on 4 September 1986) to investigate the sensitivity of model solutions to initialization start time and model spin up duration. All of the WRF simulations ended at 13 hours after the end of the tracer release (i.e., at 12:00 UTC on 5 September 1986). The three reanalysis variations included are the North American Regional Reanalysis (NARR) data (Mesinger et al., 2006), European Centre for Medium-Range Weather Forecasts (ECMWF) data (Hersbach et al., 2015), and Climate Forecast System Reanalysis (CFSR) fields (Saha et al., 2010). NARR reanalysis meteorological data are





available every 3 hours on 30 vertical levels with a horizontal grid spacing of 32 km. Both ECMWF and CFSR reanalysis fields are available every 6 hours on 38 vertical levels. However, CFSR reanalysis fields have a horizontal grid spacing of roughly 60 km versus roughly 125 km for ECMWF data.

The FDDA weighting of meteorological data fields during the WRF ensemble simulations was varied to account for un-
certainty associated with the assimilation of gridded reanalysis fields and irregularly spaced weather observations. Weather simulations were performed with WRF FDDA options for analysis and observational nudging options either turned off, using default weighting factors as suggested by WRF guidance, or a high option with the weighting factors one order of magnitude higher than the default values. Additionally, FDDA analysis nudging was used only on the two outer course-resolution model domains, while FDDA observational nudging was used only on the two innermost model domains (see WRF domains in
Fig. 3). FDDA observation nudging included surface METAR measurements and multi-level data from the backup and primary meteorological towers at Diablo Canyon (see Sect. 4).

Three PBL models and three LSM schemes were used to construct the WRF ensemble to account for uncertainty associated with turbulent mixing and surface momentum, moisture, and thermodynamic fluxes. The PBL models included the Yonsei University (YSU) scheme (Hong et al., 2006), the Mellor-Yamada-Janjic (MYJ) scheme (Janjić, 1994), and the Mellor-
Yamada-Nakanishi and Niino (MYNN) scheme (Nakanishi and Niino, 2006). Among the PBL models, the biggest difference is that the YSU scheme uses a countergradient flux (non-local) method to develop parabolic mixing profiles in the boundary layer, while the MYJ and MYNN schmes use different numerical approaches to solve for local turbulent kinetic energy (TKE) based vertical mixing in the PBL and free atmosphere. The LSM physics options include Thermal Diffusion (Duhdia, 1996), NOAH (Ek et al., 2003), and RUC (Benjamin et al., 2004) models. Soil moisture and explicit vegetation canopy physics are
not included in the Thermal Diffusion model, while the NOAH and RUC models parameterize vegetation canopy effects to differing degrees and both provide soil moisture gradients.

## 2.2 FLEXPART

The FLEXPART Lagrangian dispersion particle model (Stohl et al., 1998, 2005; Stohl and Thomson, 1999) was used to simulate the atmospheric transport and mixing of the tracer gas released from the Diablo Canyon nuclear power plant. Field
experiments have been used to validate the performance of FLEXPART (Stohl et al., 1998; Forster et al., 2007). FLEXPART has also been used in a wide variety of dispersion applications, including the transport of air pollutants (An et al., 2014; Avey et al., 2007), of radiological releases from nuclear power plants and radioisotope production facilities (Andreev et al., 1998; Wotawa et al., 2010; Stohl et al., 2012), of volcanic plumes (Stohl et al., 2011), and of noble gases produced from nuclear weapons tests (Becker et al., 2010).
We used FLEXPART-WRF version 3.1 (Brioude et al., 2013a), which was developed to use meteorological data generated by the WRF model to drive atmospheric transport and diffusion processes. The FLEXPART-WRF code is open source and available for download (https://www.flexpart.eu/). Mean particle trajectories and tracer concentrations were calculated using the three dimensional wind components from WRF ($u$, $v$, and $w$) over the 50 km by 50 km domain shown in Fig. 3 (dashed rectangular area). The FLEXPART-WRF grid used 401 cells in each of the horizontal directions and 11 vertical levels from the



surface to 3 km, with 6 levels contained in the lowest 500 m. Tracer concentrations were derived using one million particles released from a randomly selected point source location at random release times, as detailed in the next section. Wind fluctuations ($\sigma_v$ and $\sigma_w$) were calculated using parameterizations (Hanna, 1982; Ryall and Maryon, 1998) and WRF micrometeorological output variables (friction velocity, surface sensible heat flux, planetary boundary layer height, and Monin Obukhov length

scale). Lagrangian particles were evolved using a sampling rate and synchronization interval of 20 seconds, and the simulations utilized a subgrid terrain parameterization. FLEXPART-WRF can also simulate wet and dry deposition removal processes (Wesely and Hicks, 1977; McMahon and Denison, 1979; Slinn, 1982; Hertel et al., 1995), though these processes were not needed for the passive gas tracer release at Diablo Canyon.

### 2.2.1   FLEXPART Ensemble

In addition to the wind field variations generated by the WRF ensemble, the inverse modeling system in Fig. 1 also applies a Monte Carlo sampling loop to emissions variables in FLEXPART. The goal of this part of the inversion is to determine the location, timing, and magnitude of the tracer release emissions by minimizing the differences between FLEXPART predictions and field measurements.

The location, timing, and magnitude of the Diablo Canyon release are inferred by sampling the six emissions inputs shown

in Table 2. Each input is represented by a continuous random variable that can take any value in the inversion range, including the minimum and maximum values. The ranges bound the actual values used for the Diablo Canyon tracer release experiment, which are also listed in the table. The release latitude and longitude are sampled over a roughly 2 km × 2 km bounding box centered on the actual location. The height of the release is varied between 1 and 10 meters above ground, with the actual height at 2 m above the surface. Potential release start times within a two hour period centered around the actual start time

(08:00 local time) are considered. Similarly, a range of possible release durations lasting between 6 to 10 hours are tested, with the actual release occurring for 8 hours. Lastly, the inversion algorithm considered any amount between 10 and 1000 kg for the the trace gas released, with the true value at 146 kg.

The FLEXPART ensemble contains 40,000 dispersion simulations that were run and analyzed for the Diablo Canyon release. The ensemble FLEXPART simulations were generated by randomly sampling both the WRF ensemble and the FLEXPART

emissions variables. Random samples were drawn using a Latin hypercube design (Helton and Davis, 2003), which is a space-filling variation of Monte Carlo, assuming an 11-dimensional uniform probability distribution. Further details on the statistical aspects of the ensemble modeling are given below in Sect. 3.

## 3   Statistical analysis of ensembles

### 3.1   Machine learning

Machine learning is used to train statistical regression functions to approximate the input-output relationships in the WRF-FLEXPART ensemble. Once trained, the machine learning functions can be evaluated very efficiently at new input values





and used for uncertainty propagation, parameter estimation, Bayesian inference, and other types of statistical analysis. These functions are used for two primary purposes for our work. They are used to identify and rank the effects of input features in WRF and FLEXPART on the tracer responses (i.e., a form of sensitivity analysis) and to determine the values of the inputs that yield responses that are similar to tracer observations (i.e., optimization and inverse modeling). These applications are

described in more detail below.

The WRF-FLEXPART ensemble is mathematically expressed as

$$\mathbf{y} = F(\mathbf{x}_{WRF}, \mathbf{x}_{FLX}), \tag{1}$$

where $\mathbf{y}$ is a vector containing information about the output or response of the simulations in the ensemble, $\mathbf{x}_{WRF}$ and $\mathbf{x}_{FLX}$ are vectors containing the corresponding categorical and continuous inputs to WRF and FLEXPART, respectively, and the function

$F$ represents the WRF and FLEXPART physical models. The response vector $\mathbf{y}$ is taken as either the tracer concentration at a specified location and time or a measure of the goodness-of-fit between the ensemble simulations and measurements. Complex spatiotemporal dispersion patterns are not contained in $\mathbf{y}$, although new statistical methods are being developed to capture these effects (Francom et al., 2016). Machine learning is used to approximate Eq. (1) by training on the ensemble data, which results in

$$\hat{\mathbf{y}} = F(\mathbf{x}_{WRF}, \mathbf{x}_{FLX}) + \epsilon_{\mathbf{y}}, \tag{2}$$

where $\hat{\mathbf{y}}$ is an approximation of $\mathbf{y}$ and $\epsilon_{\mathbf{y}}$ is the approximation error. The value of $\epsilon_{\mathbf{y}}$ is small for our analysis and is neglected for the remaining discussion (i.e., differences between $\mathbf{y}$ and $\hat{\mathbf{y}}$ are less than 10% on average, not shown). For notational convenience, the inputs $\mathbf{x}_{WRF}$ and $\mathbf{x}_{FLX}$ are also combined into a single input vector $\mathbf{x}$ in subsequent discussion.

We use a method called gradient boosting (GB) that fits statistical regressions to Eq. (1) using sums of decision trees

following the basic notion that an individual decision tree by itself is a *weak learner*, but a combination of trees is a *strong learner* capable of fitting complex systems. The GB algorithm is briefly outlined below, and further details are available (e.g., Hastie et al., 2009). We use the GB version available in the Scikit-learn package (Pedregosa et al., 2011).

As noted, a GB model is a sum of decision trees of the form

$$F(\mathbf{x}) = \sum_{m=1}^{M} T_m(\mathbf{x};\theta), \tag{3}$$

where $T_m(\mathbf{x};\theta)$ is the tree at stage $m$ and $\theta$ is a set of tree fitting parameters (e.g., depth of the trees). An individual tree is described by

$$T(\mathbf{x};\theta) = \sum_{j=1}^{J} \gamma_j I(\mathbf{x} \in R_j), \tag{4}$$

which partitions the input space $\mathbf{x}$ into $J$ disjoint regions and assigns a value of $\gamma_j$ to region $R_j$ via the indicator function $I$. Starting from an initial model that fits the mean of the response, $F_0$, the regression model is built-up additively using a *boosting*

technique that fits trees to the residuals between the current and previous stages,

$$F_m(\mathbf{x}) = F_{m-1}(\mathbf{x}) + T_m(\mathbf{x}), \tag{5}$$





where the tree output values, $\gamma_{jm}$, are defined implicitly. This expression is solved by numerically estimating gradients of a loss function (e.g. least squares) and using steepest descent optimization. A *stochastic* variation of GB is used that considers a random subset of training data during each stage, which has been shown to improve the accuracy of the fits.

Although other statistical regression methods could be used, GB offers two clear advantages for fitting the WRF-FLEXPART ensemble. First, as shown in Eq. (2), GB naturally handles heterogeneous inputs (i.e., $\mathbf{x}_{WRF}$ and $\mathbf{x}_{FLX}$). This makes it convenient for analyzing the combined effects of WRF inputs that vary as categories or discrete variables and FLEXPART variations that vary continuously. In addition, GB has a built-in technique for determining the influence of the inputs on the outputs during training using *feature importance*. The algorithm estimates feature scores for the inputs based on the positions of the nodes in the decision trees. Inputs near the top root node affect the output more strongly than inputs near the bottom terminal nodes, and

therefore have higher feature scores. Applied to the WRF-FLEXPART ensemble, the feature scores are analogous to sensitivity coefficients that quantify the fraction of the ensemble variance caused by the changes in the inputs.

### 3.2   Bayesian inverse modeling

The goal of the inverse modeling is to determine the values of the inputs to WRF and FLEXPART that provide output concentrations that best match the tracer measurements. The inversion uses an extension of our approximate Bayesian computation

algorithm described in Lucas et al. (2016). The algorithm has been updated to enable joint inversions of categorical inputs in WRF and continuous inputs in FLEXPART, and to allow more flexibility in the calculation of the model-observation likelihood distance weights. In particular, the likelihood weights now utilize predictions from the GB regressions and consider more than one distance metric. Further information about the scheme is provided below and is illustrated in Fig. 5.

    The inverse method applies Bayes' rule,

$P(\mathbf{x}|\mathbf{y}) \propto P(\mathbf{y}|\mathbf{x})P(\mathbf{x}),$                               (6)

to estimate $P(\mathbf{x}|\mathbf{y})$, which is the conditional probability density function of the WRF and FLEXPART inputs, $\mathbf{x}$, given simulations and measurements of tracer concentrations, $\mathbf{y}$. The prior probability distribution of model inputs, $P(\mathbf{x})$, is an 11-dimensional uniform probability distribution over the WRF and FLEXPART random variables listed in Tables 1 and 2. As illustrated on the left hand side of Fig. 5, the prior distribution uses uniform categorical random variables for the WRF inputs

and uniform continuous random variables for the FLEXPART inputs. Samples are drawn from the prior distribution using a Latin hypercube method (Helton and Davis, 2003).

    The remaining term in Eq. (6) is the likelihood function, $L = P(\mathbf{y}|\mathbf{x})$, which quantifies the level of agreement between the simulated and measured tracer concentrations for a given draw from the prior distribution. Relatively high and low likelihood values correspond, respectively, to simulations that agree well and poorly with measurements. Following our previous work

(Lucas et al., 2016), we compute the mean-squared-error (*mse*) between simulations and measurements as one metric for calculating high and low likelihoods. Furthermore, we include the correlation (*corr*) between observations and simulations as another metric to determine high and low likelihood values. The *corr* metric is included because it is sensitive to different aspects of model and observation differences than the *mse*. The *mse* varies with the magnitude of the differences between





observations and simulations, and is expected to be mainly sensitive to changes in the source amount. The temporal correlation, on the other hand, is expected to be sensitive to changes in the arrival time and duration of the plume at the measurement locations. By combining the two metrics *mse* and *corr* into a single likelihood weight, we aim to constrain a larger number of input parameters than we could using either metric alone.

To account for the two metrics in the likelihood function, we use an expression of the form

$$\log L = -0.5 [\mathbf{y_s}(\mathbf{x}) - \mathbf{y_t}]^\mathsf{T} \mathbf{\Sigma}^{-1} [\mathbf{y_s}(\mathbf{x}) - \mathbf{y_t}], \tag{7}$$

where $\mathbf{y_s}$ is a column vector of the metrics for a simulation at input $\mathbf{x}$,

$$\mathbf{y_s}(\mathbf{x}) = \begin{pmatrix} \mathrm{mse} \\ \mathrm{corr} \end{pmatrix}, \tag{8}$$

$\mathbf{y_t}$ is the corresponding column vector of the metric "targets", and $\mathbf{\Sigma}$ is the $2 \times 2$ covariance matrix of model and observation

errors. The highest likelihood values occur at the inputs that jointly minimize the *mse* and maximize the *corr*. Ideally, with a perfect model and data, the targets for *mse* and *corr* would be 0 and 1, respectively. In practice, however, models are imperfect and data is noisy, so it is usually not possible to find simulations that match the data perfectly. To avoid extrapolation, we therefore define the targets in $\mathbf{y_t}$ using a small number of the best fitting simulations within the ensemble, and then estimate the covariance matrix $\mathbf{\Sigma}$ using a bootstrap resampling procedure (Wilks, 2011). Further details of the target and covariance

estimation are provided in Sect. 5.4.

  Before computing the *mse* and *corr* in Eq. (8), the tracer concentrations are transformed using a Box-Cox power transformation with an exponent of $-0.25$ (see Wilks, 2011). This transformation generalizes the logarithmic transform and is used because the tracer concentrations vary over many orders of magnitude. Without it, the likelihood metrics would be skewed toward higher tracer concentrations near the release location. The Box-Cox transformation also symmetrizes the distribution

of differences in Eq. (7) by removing long tails.

  The Bayesian inversion is performed using GB regressions, instead of actual model simulations, to predict $\mathbf{y_s}(\mathbf{x})$ for 2 million new Latin hypercube input values. Two million points are needed to better cover the large sampling volume of the 11-dimensional prior distribution because the sampling volume varies exponentially with the number of dimensions. For instance, partitioning the ranges of only the 6 FLEXPART inputs into 10 bins each results in a volume with $10^6$ bins. Running and

analyzing the output of $10^6$ FLEXPART simulations is computationally infeasible, so we use the ensemble of $10^4$ simulations (Sect. 2) as a training dataset to build the GB regressions, and then use the regressions as surrogates for the actual models in the inversion because they can be evaluated very efficiently.

  To verify the Bayesian inversion scheme, we performed a series of "synthetic data" tests using model-generated inputs and outputs. These tests are important because inverse problems often have multiple solutions and may be poorly constrained (i.e.,

ill-posed and ill-conditioned). Sect. 5.5 highlights the results of a synthetic data inversion test.





## 4 Diablo Canyon tracer release experiment

Field measurements from the Diablo Canyon nuclear power plant tracer release experiment (Thuillier, 1992) are compared to the ensemble simulations and used to test the Bayesian inversion algorithm. The Pacific Gas and Electric Company (PG&E) owns and operates Diablo Canyon and conducted the tracer experiment in 1986 to evaluate and improve PG&E's dispersion

modeling capabilities in case of an accidental release. Figures 2 and 3 show the geographical setting around Diablo Canyon, which is located on the California coast in complex terrain near San Luis Obispo. The plant sits on a shelf about 26 m above sea level and is surrounded by hills with peaks about 500 m above sea level and many canyons. The hills block onshore, westerly flow, which creates challenges in simulating the effects of plumes released from Diablo Canyon on the population centers of San Luis Obispo and Pismo Beach.

PG&E conducted eight tracer release tests between August 31 and September 17, 1986. Although the large-scale wind patterns for the eight tests showed relatively similar onshore flow from the northwest, the third tracer test on September 4 experienced a strong seabreeze that presents a challenge for dispersion modeling. We therefore use the third tracer release for our inversion testing. Starting at 8 AM Pacific Daylight Time (PDT) on September 4 (15:00 UTC), 146 kg of the passive tracer sulfur hexafluoride ($SF_6$) was released 2-m above ground from Diablo Canyon steadily over an 8-hour period (8 AM to 4 PM

PDT). The concentration of $SF_6$ was measured at the network of 150 tracer air sampling locations shown in Fig. 3 (black dots).

The measurement network was designed to monitor the expected tracer transport paths near terrain gaps, the entrances and exits to the inland valley, and the coastal boundary. An arc of 24 sampling sites was positioned very close to the nuclear power plant release point at a radial distance of 840 m (black dots surrounding Diablo Canyon in Fig. 3). This arc was designed to detect the initial direction of transport of released plumes and to provide details of dispersion at the nominal plant boundary.

A second, linear array of 8 samplers was placed about 7 km southeast of the plant to detect transport and indicate dispersion characteristics along the anticipated principal coastal transport path for plumes released at Diablo Canyon (i.e., see site 330 in Fig. 3). Tracer sampling was done automatically by sequential pumps filling polyvinyl fluoride bags. An integrated sample was taken over each hour at each sampling site from 7 AM to 7 PM PDT. This allowed for one-hour samples prior to the tracer release for the purpose of estimating tracer background levels, and three one-hour samples after the cessation of release for the

purpose of following the tracer as it traveled through the domain.

Figure 4 shows the $SF_6$ measurements used for the inversion. Of the 150 locations in the Diablo Canyon tracer release measurement network, 137 stations are contained in the FLEXPART domain represented by dashed rectangle in Fig. 3. The left hand side of Fig. 4 shows the pre-release $SF_6$ concentrations measured at these 137 stations labeled by their site identification numbers. The pre-release concentrations are used to gauge background values of $SF_6$. As shown, there are a handful of site 300

stations that have highly elevated values of $SF_6$ (above 100 ng $m^{-3}$). These values are well above background tropospheric $SF_6$ levels (Rigby et al., 2010), and are therefore due to local pollution. Sites 312–323 are affected by local pollution and are contained within the southern portion of the 840 m arc of stations surrounding Diablo Canyon. The source of the pollution is likely due to fugitive emissions from a power switchyard located within the Diablo Canyon premises. Rather than attempt to account for the extra source from the fugitive emissions, we instead exclude sites 312–323 from our analysis. After removing





these sites, a dataset with 1,148 one-hour averages of the $SF_6$ concentration measured at 125 stations is used for the inversion (352 points are missing in the raw data). The resulting distribution of $SF_6$ concentrations is shown on the right hand side of Fig. 4.

## 5 Results

### 5.1 Forward model plumes

Examples of simulated 30-minute average dispersion plumes are shown in Fig. 6. These simulations use identical input settings and parameters in WRF and FLEXPART, except for the reanalysis fields. The plumes on the left use NARR and those on the right use ECMWF. The remaining WRF settings follow the base case values listed in Table 1, while the FLEXPART simulations use the actual values of the source release parameters listed in Table 2. These plumes therefore represent the our best prior knowledge in a forward modeling sense, and provide tracer concentrations that we would expect to compare reasonably well to measurements without inverse modeling.

The upper portion of the figure shows the plumes using NARR and ECMWF five hours after the release. At this stage of the simulations, there is a large spatial difference between the plumes. The dispersion using NARR is directed eastward, is spatially more confined, and does not extend downwind of Pismo Beach, as compared to the southeast directed ECMWF plume. The ECMWF plume covers a much wider region, though most of the extended area is over the ocean. Because there are not many measurement sensors over the ocean, we expect there to be smaller differences between NARR and ECMWF in the inversion algorithm than the plumes in upper part of Fig. 6 suggest.

Nine hours after the release, as shown in the lower part of the figure, the plumes using NARR and ECMWF begin to resemble each other. Both are directed to the southeast, and both have about the same spatial extent. The higher concentration area of the plumes using ECMWF are a little more dispersed near the release location (see red contour), but otherwise the differences between the two reanalysis cases are minor. The next section shows that differences are substantial when the 11 model inputs are varied together.

### 5.2 Prior probability distribution of $SF_6$

Time series of the ensemble of $SF_6$ concentrations at four representative Diablo Canyon tracer measurement sites are displayed in Fig. 7. These time series, which are from the 40,000 member Latin hypercube ensemble, provide an estimate of the prior probability distribution of $SF_6$ concentrations due to variations in meteorology and emissions because they sample the uniform random variables in WRF and FLEXPART without considering observational constraints. The time series at each site displays the reference WRF-FLEXPART base simulation (black line), the tracer measurements (red squares), and different quantiles of the prior $SF_6$ distribution, including the median (solid blue line) and 5-95% range (light blue shading).

Starting with the time series at Site 325, which is closest to the release point, the simulated $SF_6$ concentrations are negligible until about 9 AM, when the plume begins to pass over the location. The concentrations stay elevated for about 8 hours and





then drop off as the trailing edge of the plume moves over the site. The time series at the other sites show similar behavior, except that the arrival times of the plume are delayed and the peak concentrations are reduced relative to their distance from the release point. Considering the large 5-95% quantile range in the time series, which spans about three orders of magnitude in the $SF_6$ concentrations, the tracer measurements generally agree with the ensemble, other than during the initial 2–3 hours.

Also note that, although the reference case appears to agree with the measurements reasonably well (i.e., the black versus red lines), the inversion algorithm may be able to identify a WRF configuration that provides a better match.

### 5.3    Feature scores of $SF_6$

The $SF_6$ concentrations in the prior probability distribution in Fig. 7 vary by three orders of magnitude due to variations in the inputs to WRF and FLEXPART. Gradient boosting tree regressions are used to estimate the input feature scores, where

the score for a given input is analogous to the sensitivity index quantifying the fraction of the variance caused by changes in that input. The feature scores are extracted from fitting individual GBTRs to the Latin hypercube ensemble at each site and for 30-minute concentration average periods. We only fit GBTRs during the periods when there is significant plume ensemble variability present at all of the locations simultaneously, which occurs between 12:00 and 20:00 PM local time.

Figure 8 displays the resulting time series of the GB-based feature scores at the four representative sites. The stacked color-

coded bands show the fraction of ensemble variance in the prior distribution explained by the 11 inputs, with the scores for the FLEXPART inputs at the bottom and the WRF inputs at the top of each stack. The patterns are generally similar at different locations and times, which indicates that there is not a strong spatial or temporal component to the feature scores or model sensitivities. The patterns also show that the FLEXPART and WRF inputs cumulatively account for about 80% and 20%, respectively, of the ensemble variance in Fig. 7. On the FLEXPART input side, the latitude and longitude of the release and

the source amount are the most important features explaining the variance of the prior probability distribution. Among the WRF inputs, the feature scores associated with different reanalysis fields are slightly higher than the scores of the other WRF features.

Overall, the feature scores suggest that the prior uncertainty in the source term inputs are more critical than the prior uncertainty in the meteorological inputs for this particular tracer release experiment. Although the WRF inputs are not the dominant

source of variability, the combined effects of the sources of meteorological uncertainty still cannot be neglected. It is also important to note that, for tracer release simulations conducted under different meteorological conditions, at different space and time scales, or that consider additional sources of uncertainty, the input sensitivities will likely differ from those estimated here. Moreover, the contribution of meteorological uncertainty is expected to be larger for forecast problems that are not constrained by reanalysis data.

In addition to being useful for understanding the drivers of variance in the prior probability distribution, the feature scores are also useful for interpreting the results of the Bayesian inversion in the following sections. Inputs with relatively high feature scores are often easier to constrain with observations. On this basis, therefore, we expect the posterior probability distributions for the FLEXPART location and source amount inputs to be relatively narrower than the other FLEXPART terms because they have the highest feature scores.





### 5.4  Likelihood distance metrics

Figure 9 displays the mean-squared-error and correlation between the ensemble simulations and the tracer measurements. The blue and red dots in the figure show the *mse* and *corr* for the 40,000 individual FLEXPART simulations, where for each simulation the metrics are computed using 1,148 hourly-average $SF_6$ concentration measurements collected at 125 locations within the simulation domain over the 12 hour period (352 measurements are missing). The simulations that provide a reasonable fit to the measurements have low values of *mse* and high values of *corr* and are located in the upper left quadrant of the figure, while those that disagree with the measurements are located in the lower right portion of the figure. The 50 best matching simulations are displayed using red dots in the upper left. As noted earlier, the best values of *corr* and *mse* in the figure are far from the perfect values of 1 and 0, respectively, because the WRF and FLEXPART models are imperfect and the measurements are noisy.

The points in the figure are used to estimate the terms in the likelihood function in Eq. (7) by the following procedure. First, the data points are used to form a training dataset to fit GB regressions to predict the *mse* and *corr* at new simulation input values that are not part of the ensemble. The resulting GB models fit the data very well (not shown), having coefficients of determination between actual and predicted values of $R^2$ = 93% and 98% for *mse* and *corr*, respectively. New Latin hypercube samples are then drawn and evaluated in the GB models to form the simulation vectors $\mathbf{y_s}(\mathbf{x})$ in Eq. (7). The remaining terms, the target vector $\mathbf{y_t}$ and covariance matrix $\mathbf{\Sigma}$, are estimated from the best fitting simulations in the figure (i.e., the red dots) by applying a bootstrap technique that resamples the tracer measurements with replacement (Wilks, 2011). We loop through the set of best fitting simulations 500 times, and each time use a random subset of the measurements (50%) to recompute *mse* and *corr*. The target vector and covariance matrix are then estimated by fitting a multivariate Gaussian distribution to the bootstrapped points, yielding the mean and standard deviation ellipses shown in the figure. The distribution is used in the inversion to calculate the posterior probability distributions of the model input values (Sect. 5.6) and tracer concentrations (Sect. 5.6).

### 5.5  Inversion with synthetic data

Before performing an inversion with the Diablo Canyon tracer data, we first apply the algorithm to "synthetic" data with known inputs and outputs as a verification test. Synthetic data is generated using the output concentrations from a randomly selected ensemble member as the inversion target. The posterior probability distribution of parameter values is computed using the previously described methods (i.e., fitting GB regressions to the *mse* and *corr* and estimating the covariance matrix as in Sect. 5.4). We draw 2 million new Latin hypercube points from the prior distribution to better cover the 11-dimensional input space, evaluate these points in the likelihood function, and compare the maximum likelihood locations to the known input values.

Figure 10 shows an example of a synthetic data inversion test using an arbitrarily selected Latin hypercube run. Other simulations have also been tested, but the results are not displayed here. The figure shows the posterior distribution of model parameter values. The FLEXPART parameters are displayed in the left hand portion of the figure using continuous distributions,





while the WRF parameters are shown on the right using categorical distributions. The plots along the diagonal show univariate marginal distributions for the labeled parameters with the vertical axis indicating the normalized probability density. The off-diagonal plots show bivariate marginal distributions for the pair of parameters in the corresponding row and column with the red colors showing regions of high probability density. The known input values are denoted by the black vertical lines and circles.

For this particular test, the WRF simulation used the 06:00 UTC initialization time, the NARR reanalysis data, the MYNN TKE PBL scheme, the RUC land surface model, and no data assimilation nudging. The Bayesian inversion algorithm successfully determines these inputs, because the areas of highest posterior probability density coincide with the known values (i.e., the tallest bars and red areas overlap with the black lines and circles). All of the WRF inputs, but the land surface model type, exhibit large differences across the posterior categories, which indicates that the inputs are well constrained by the data and metrics. In particular, there is little to no posterior weight associated with the other initialization time and reanalysis fields. The relatively small differences across the LSM categories are thought to occur because the plume is predominantly transported over the ocean and sampled near the coast (see Figs. 3 and 6), and therefore the *mse* and *corr* metrics are not sensitive to changes in the land surface model.

The positions of the peak values in the posterior distribution for the FLEXPART dispersion inputs also agree exceedingly well with the known input values. Except for the release altitude, the algorithm infers the location, amount, and timing of the source. As determined from the widths of the posteriors, the release latitude and longitude are the best constrained FLEXPART inputs, followed by the source amount and duration. The posterior distribution for the release altitude is relatively unchanged from the flat prior distribution, suggesting that the data and metrics are not sufficient to constrain the relatively small variations of the release altitude (0 to 10 meters). As previously noted, there is a reasonable correspondence between the widths of the FLEXPART posterior widths in Fig. 10 and the size of the feature scores in Fig. 8.

These results, along with other synthetic data tests that are not shown, provide confidence that the inversion algorithm appears to be functioning adequately. The algorithm returns values for the WRF and FLEXPART inputs that are close to the actual values for most of the parameters.

## 5.6 Inversion with Diablo Canyon tracer data

For the inversion using the $SF_6$ measurements, we draw another 2 million Latin hypercube points from the prior distribution for WRF and FLEXPART, evaluate the points in the GB fits for *mse* and *corr*, and then compute the likelihood weights relative to the target and covariance displayed in Fig. 9. The resulting posterior distribution of WRF and FLEXPART parameters is shown in Fig. 11.

As is the case with the synthetic inversion tests, the actual values for the location, start, duration, and amount of the $SF_6$ release are known for this tracer experiment. The inversion, however, assumes that the release parameters are unknown and uses the measurements to infer their values. As shown in the figure, the data and algorithm are sufficient to determine most of the FLEXPART source term parameters, because the positions of the maximum likelihood values of the parameters closely match the known experimental values (black lines and circles in Fig. 11). The close agreement between the two implies that the





WRF and FLEXPART models do not have any severe deficiencies that prevent them from accurately simulating tracer transport for this experiment. The synthetic data inversion tests from the previous section would not expose model deficiencies because the same deficiency would be present in both the simulations and target, and hence would be subtracted out of the analysis.

Referring to Fig. 11, we see that the marginal distributions for the latitude, longitude, and amount of the release have the

sharpest peaks and are therefore the most constrained by the measurements. The source start and duration are also moderately constrained, though the distribution for release height is unconstrained and remains essentially flat. The posterior distribution also suggests that the release duration lasts longer than the 8 hour period used in the experiment. It is difficult to simulate rapid changes associated with the leading and trailing edges of the plume, and so the model and methods may be smoothing out these features and causing the overestimation of the duration. Other likelihood metrics besides *mse* and *corr* may help alleviate this

issue.

The posterior distribution also shows a strong covariance relationship between the release latitude and longitude (see bivariate distribution in the upper left of Fig. 11). An area of relatively high probability stretches from northwest of the actual release location to the southeast. The shape of this covariance stems from the large-scale flow pattern and nearby measurement locations. The general direction of the flow for the release period is from the northwest to the southeast, and the release point

is situated within a fairly close arc of sensors (see Fig. 3). As long as the release stays within this arc, moving the release location slightly upwind or downwind will not greatly affect the simulated concentrations at the sensor locations. If the release location is moved orthogonal to the flow or outside of the arc, however, FLEXPART will simulate $SF_6$ at sensors where none was measured, and vice versa.

The WRF configurations in the posterior distribution are displayed on the right hand side of Fig. 11. Unlike the known

FLEXPART inputs described above or the known inputs in the synthetic data experiments, the actual values of the meteorology are not known here (i.e., there are no black lines or circles for WRF). The configurations that minimize the *mse* and maximize the *corr* with the $SF_6$ measurements are represented by the tallest bars in the univariate distributions along the diagonal and by the red-colored squares or bands in the off-diagonal bivariate distributions.

As shown in the figure, the maximum likelihood configuration consists of the 06:00 UTC initialization time, the ECMWF

reanalysis fields, the YSU PBL scheme, the RUC land surface model, and no data assimilation nudging. Some of these configuration settings may, at first, seem surprising. For example, the NARR reanalysis fields have a higher spatial resolution than the ECMWF fields and therefore may be expected to perform better. Likewise, the option to run without data assimilation seems to outperform the options with assimilation. Referring to the figure for these cases, the posterior distribution still has significant probability density for both the NARR and low nudging options. Compared to the posterior distribution in the synthetic data

inversion, the WRF inputs are not as strongly constrained using the tracer data, especially the inputs for the land surface model and nudging. Only two of the WRF inputs have settings with negligible probability, the earlier initialization time and CFSR reanalysis data. The alternate PBL schemes also have relatively low probability. We therefore conclude that the winds generated using the 06:00 UTC initialization time, the NARR or ECMWF reanalysis fields, and the YSU PBL scheme will optimize our likelihood metrics, and that there is not a preferred land surface model or nudging option.



### 5.7 Posterior probability distribution of SF$_6$

The ensemble time series in Fig. 7 are based on sampling the prior distribution of input parameters, which results in a substantial spread of SF$_6$ concentrations over two to three orders of magnitude. Most of these ensemble members do not agree well with the tracer measurements, so we estimate the posterior distribution of SF$_6$ concentrations by applying the likelihood weights of

Sect. 5.4 and 5.6 to the ensemble time series in Fig. 7.

Figure 12 displays the posterior distribution of the time series of SF$_6$ concentrations. As before, the time series show the tracer measurements (red squares) and quantiles of the posterior distribution, including the median (solid blue line), first and third quartiles (dashed blue lines), and 5-95% range (light blue area). The general features of the posterior distribution are similar to the prior distribution, except that the ensemble spread has been greatly reduced and the quantiles have shifted to

higher values.

Other than at site 325, where a large spread remains, the 5-95% range covers about one order of magnitude, or a reduction of two orders of magnitude. Even with the reduced range, most of the measurements still fall within the light blue area. We do not expect all of the measurements to lie in the 5-95% range because the likelihood metrics consider the aggregation of all of the sites and times. In order to achieve an overall higher likelihood, individual measurement points may be farther away from

the median in the posterior distribution than they were in the prior.

In addition to the reduction in the variance, Fig. 12 also shows that the median and other quantiles of the posterior distribution shift to higher concentrations. This shift occurs because the source term parameter variations in the prior distribution lead to many simulations having SF$_6$ concentrations that are too low relative to the measurements. The likelihood weights discount these simulations,

## 6   Conclusions

We have developed an ensemble-based Bayesian inverse modeling system that can determine information about an atmospheric release from a nuclear power plant using measurements collected a relatively safe distance downwind from the plant. The system uses an ensemble of WRF simulations to capture uncertainty in meteorological fields and an ensemble of FLEXPART dispersion simulations to vary factors related to emissions. Machine learning algorithms are trained on the input-output rela-

tionships in the meteorological and dispersion ensemble, resulting in statistical surrogate models that mimic the behavior of the actual WRF and FLEXPART models, but that can be evaluated very rapidly at millions of new input value combinations.

Using our system, we can determine the input factors that are most important for understanding and reducing uncertainty in the ensemble (i.e., sensitivity analysis) and can estimate the values of the model inputs that provide likely matches between model output and field measurements (i.e., inverse modeling). Bayes' rule is used for the inversion, which provides probability

distribution functions of model inputs and outputs constrained by observations and that serve as a quantitative assessment of model performance. The inversion is designed to estimate the location, timing, and amount of material released to the atmosphere, and to determine the best categories of settings for running a meteorological model. The inversion system can





handle, without difficulty, additional factors related to the transport and dispersion of potential materials released during a nuclear power plant accident (e.g., wet and dry deposition of soluble radioactive products).

Our ensemble system is tested against a tracer release experiment conducted near the Diablo Canyon nuclear power plant located in the rugged terrain of coastal California (Thuillier, 1992). An ensemble of 40,000 dispersion simulations is created using a Monte Carlo method to sample uncertainty in 6 source term parameters in FLEXPART and 5 meteorological categories distributed among 162 unique configurations in WRF. The variance of the resulting unconstrained tracer concentration ensemble is substantial (i.e., the prior distribution), covering a 5% to 95% concentration probability range of about four orders of magnitude. About 80% of the unconstrained variance is due to source term parameter variations, with about half the overall variance coming from just three input parameters (release amount, latitude, and longitude). Although the meteorological inputs are not dominant sources of ensemble variability, they cumulatively account for 20% of the variance in the prior distribution and are important because their uncertainty is not easily reduced.

By calculating the mean squared error and correlation between the tracer measurements and the surrogate model predictions, the Bayesian inversion algorithm produces a posterior distribution of model inputs and outputs for the tracer release experiment. Even though the source term parameters are initially unknown in the inversion (i.e., an non-informative prior), the most likely posterior values of the FLEXPART inputs closely estimate the actual values used in the tracer release experiment, which demonstrates a successful inversion. Table 3 summarizes the results of the tracer release source inversion. As shown, the most likely values of inversion algorithm are within about 200 meters of the release location, within 5 minutes and 50 minutes of the starting time and duration, respectively, and within 22% of the actual release amount. Furthermore, the posterior values of the WRF inputs show a preference for particular configurations involving the later initialization time, YSU PBL scheme, and NARR and ECMWF reanalysis fields. Compared to the large concentration spread in the prior distribution, the posterior variance of tracer concentrations is greatly reduced and better tracks the measurements, thus indicating a good correspondence between the posterior inputs and outputs.

It is important to keep in mind that the ensemble and inversion methods can be applied to problems other than nuclear power plant releases. While the location of a nuclear power plant release is generally restricted to reactor buildings or other nearby facilities, the inversion algorithm can also determine an arbitrary release location from within a large area (e.g., 100's of km$^2$) if suitable observations are available.

*Acknowledgements.* This work was performed under the auspices of the U.S. Department of Energy by Lawrence Livermore National Laboratory under Contract DE-AC52-07NA27344 and was funded by Laboratory Directed Research and Development at LLNL under project tracking code PLS-14ERD006. The manuscript is released under UCRL number LLNL-JRNL-710162. The authors thank PG&E for access to the Diablo Canyon measurement data, and Livermore Computing for providing computational resources through an institutional allocation for the Monte Carlo simulations. The authors also thank Devin Francom and Bruno Sansó from UC Santa Cruz, and Vera Bulaevskaya from LLNL for invaluable discussions about the statistical analysis.




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





**Table 1.** Categorical Random Variables for WRF Ensemble.

| Categorical Variable | Description | Category | Label |
|---|---|---|---|
| 1. WRF_initim | Initialization time | 1986-09-04-06 | 0* |
| | | 1986-09-04-00 | 1 |
| 2. WRF_reanalysis | Reanalysis data | NARR | 0* |
| | | ECMWF | 1 |
| | | CFSR | 2 |
| 3. WRF_nudge | Nudging | Off | 0 |
| | | Low | 1* |
| | | High | 2 |
| 4. WRF_pbl | Boundary layer physics | YSU | 0 |
| | | MYJ TKE | 1* |
| | | MYNN TKE | 2 |
| 5. WRF_lsm | Land surface model | Thermal diffusion | 0 |
| | | Noah | 1* |
| | | RUC | 2 |

∗ denotes a WRF setting used for the reference base simulation.





**Table 2.** Continuous Random Variables for FLEXPART Tracer Ensemble.

| Continuous Variable | Description | Actual Value | Inversion Range |
|---|---|---|---|
| 1. FLX_loc_lat | Release latitude | 35.2111°N | [35.1977, 35.2250] |
| 2. FLX_loc_lon | Release longitude | 120.8543°W | [120.8708, 120.8384] |
| 3. FLX_zlev_bot | Release altitude | 2 meters | [1, 10] |
| 4. FLX_source_start | Release start | 08:00 local time | [07:00, 09:00] |
| 5. FLX_source_duration | Release duration | 8 hours | [6, 10] |
| 6. FLX_source_amount | Release magnitude | 146.016 kg | [10, 1000] |

**Table 3.** Most Likely Source Parameters for the Diablo Canyon Tracer Release.

| Source Term | Maximum Likelihood | Actual |
|---|---|---|
| Latitude | 35.2125°N | 35.2111°N |
| Longitude | 120.8560°W | 120.8543°W |
| Start time | 08:05 local time | 08:00 local time |
| Duration | 8.83 hours | 8 hours |
| Amount | 177.830 kg | 146.016 kg |





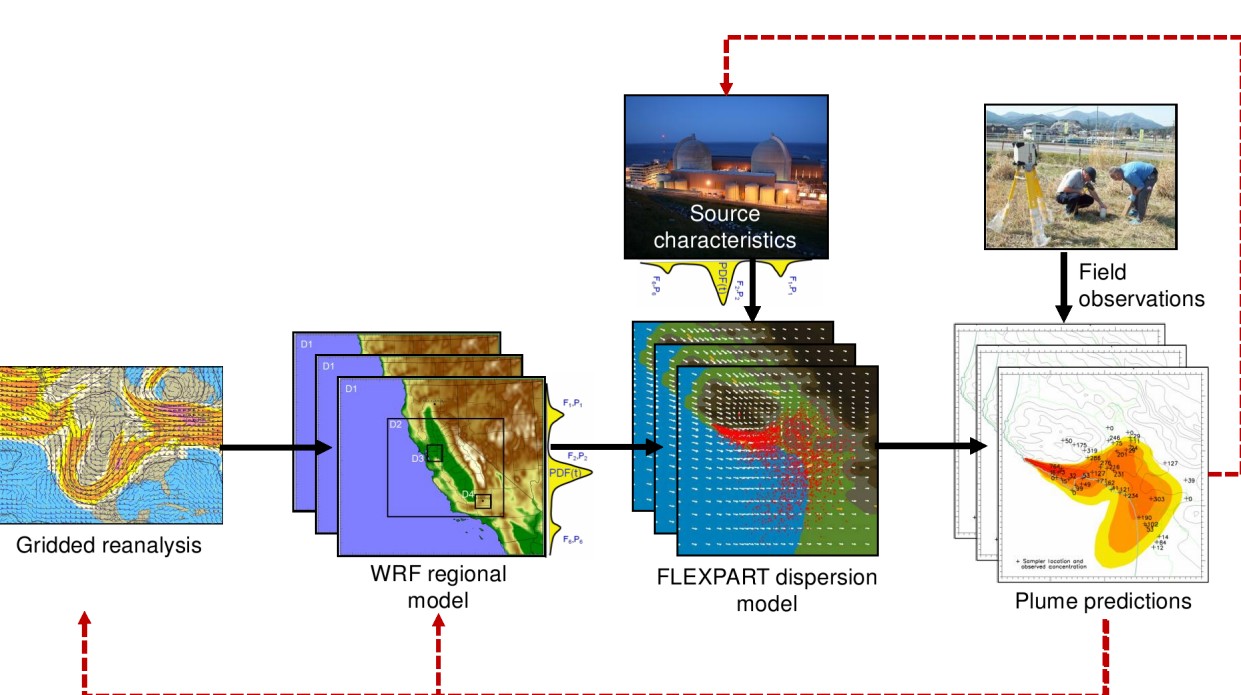

**Figure 1.** Overview of the ensemble WRF-FLEXPART system for inverse modeling and quantifying uncertainty in plume predictions due to uncertainty in meteorology and source release characteristics. The red dashed arrows show Monte Carlo sampling loops used to improve the differences between simulations and observations.





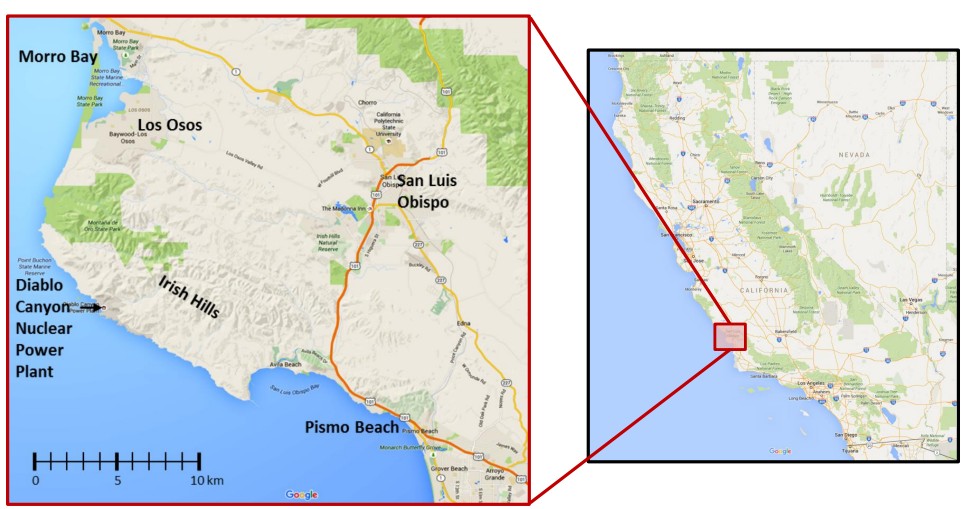

**Figure 2.** Maps showing the geographical location of the Diablo Canyon nuclear power plant on the coast of central California near San Luis Obispo and Pismo Beach.



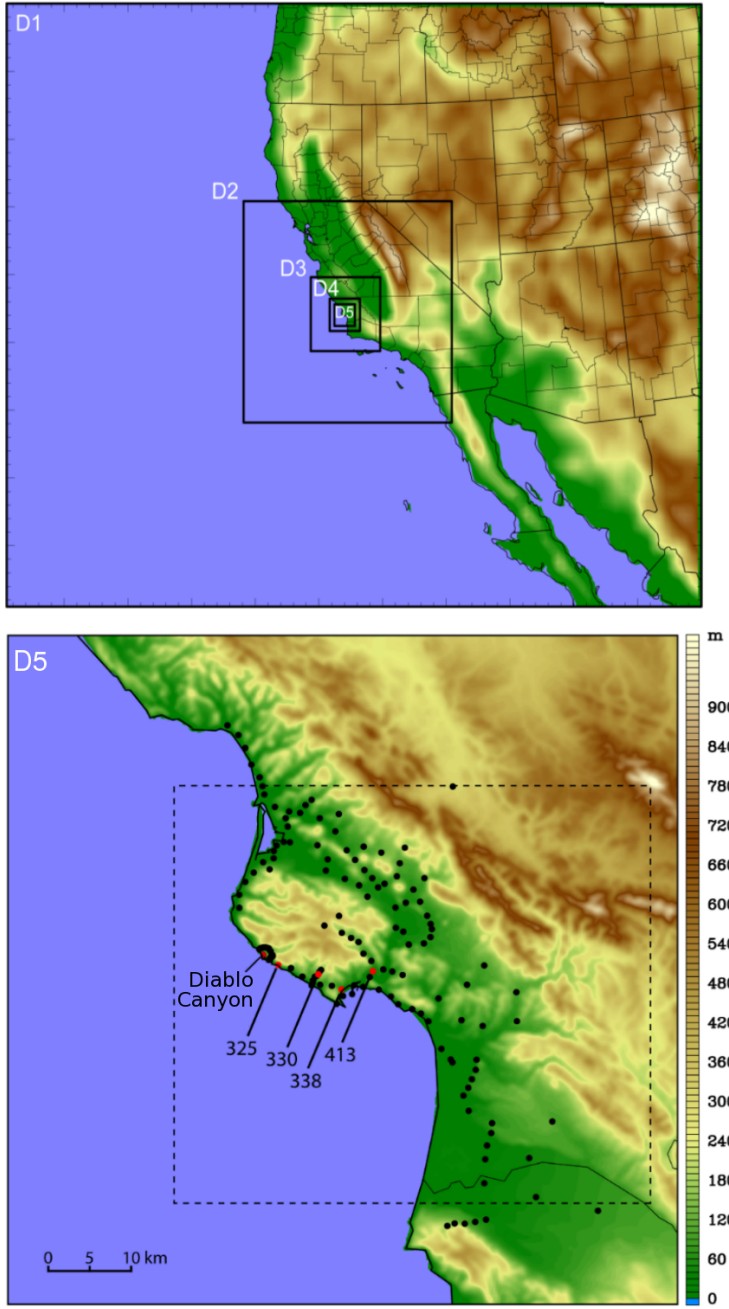

**Figure 3.** The upper panel shows the five nested domains used in WRF for simulating meteorological conditions for the Diablo Canyon nuclear power plant tracer release experiment (D1 to D5). The lower panel highlights the inner WRF domain (D5), the FLEXPART domain (dashed rectangle; longitudes of 120.954°W to 120.343°W, latitudes of 34.948°N to 35.389°N), the tracer release location (Diablo Canyon, red dot), and the measurement network (black dots) used for the tracer inversion. Four representative measurement sites (325, 330, 338, and 413) are highlighted (red dots).





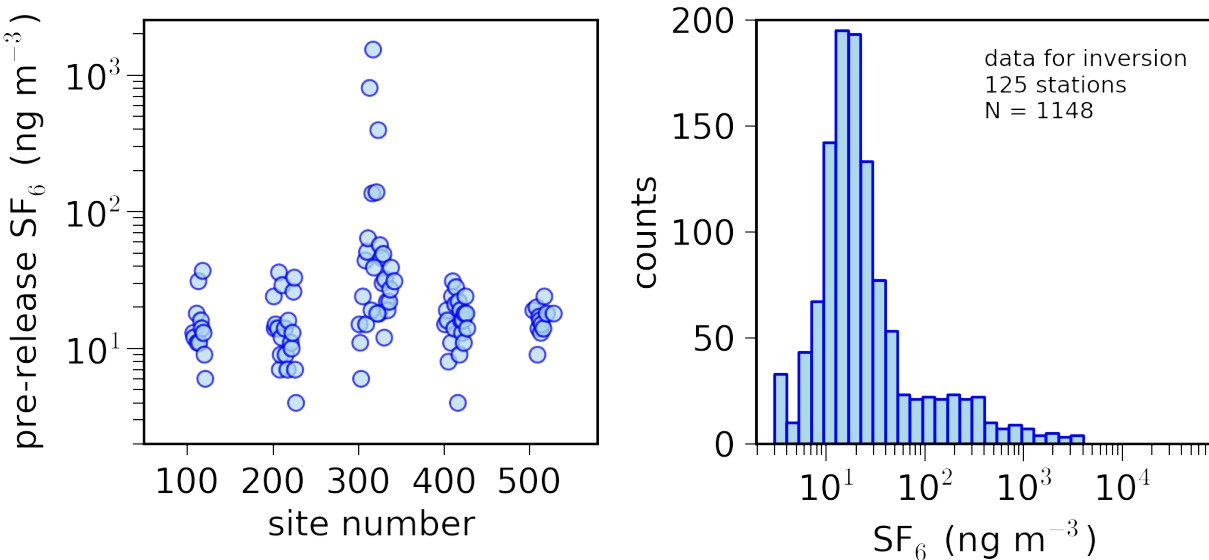

**Figure 4.** One-hour average $SF_6$ concentrations measured at the Diablo Canyon measurement sites during the tracer release test on the September 4, 1986. Pre-release concentrations (left hand side) show the effects of fugitive emissions at a subset of the site 300 locations (levels above 100 ng m$^{-3}$). Histogram of $SF_6$ concentrations from 1,148 measurements (right hand side) are used by the inversion algorithm to compute likelihood weights.





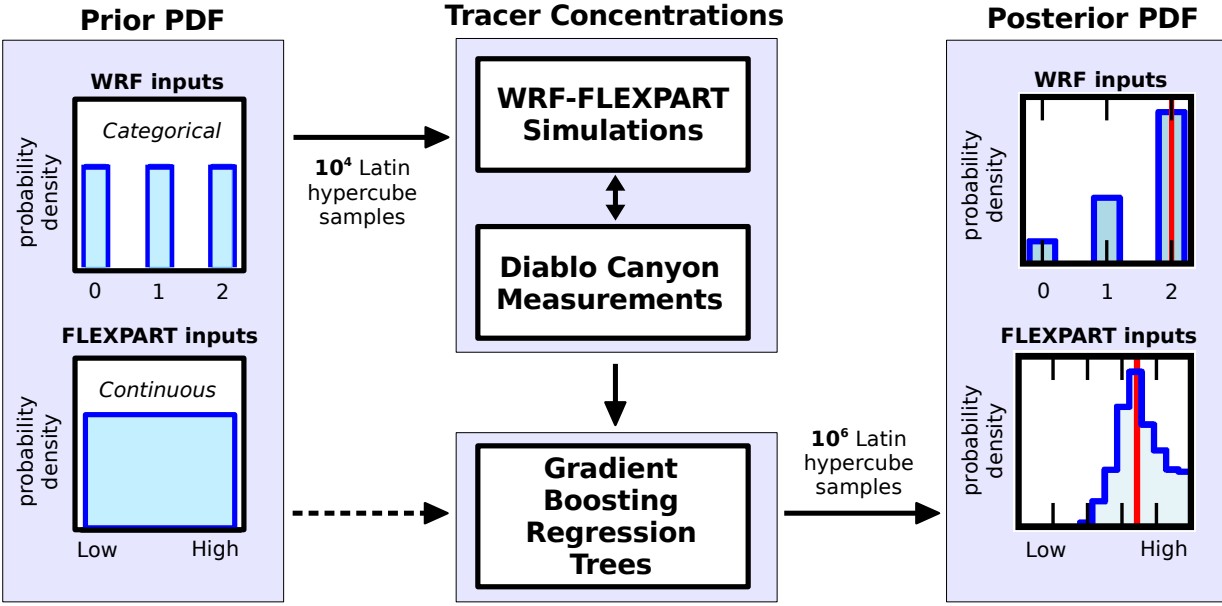

**Figure 5.** Bayesian inversion method for constraining the WRF and FLEXPART inputs. Samples are drawn from the uniform prior distribution on the left, and then evaluated in WRF-FLEXPART and compared to measurements. Gradient boosting regression trees are fit to model-measurement differences and used to infer the posterior distributions of the inputs.





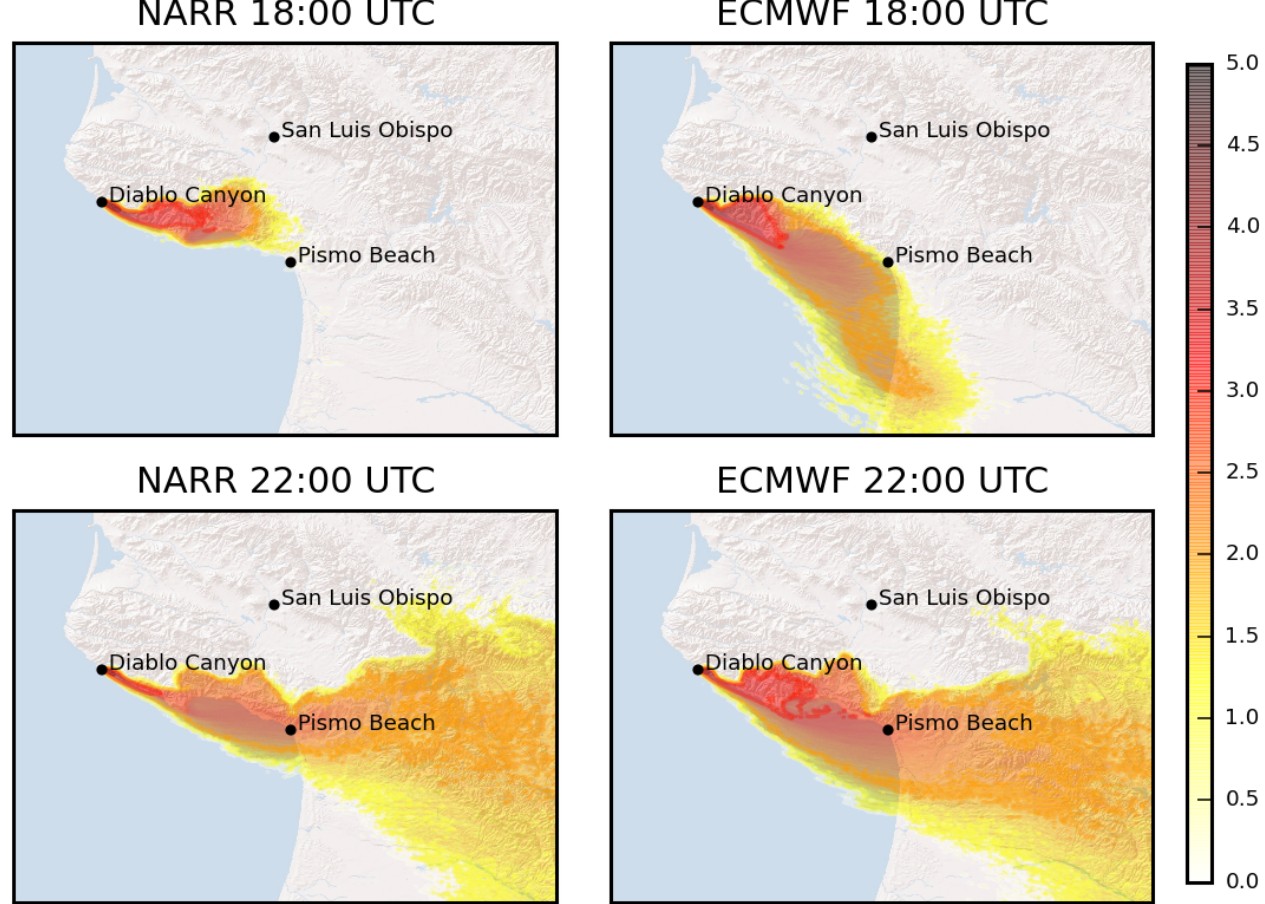

**Figure 6.** 30-minute average plumes of SF$_6$ simulated using FLEXPART with the actual release parameters (see Table 2). The plumes in the upper and lower panels are five and nine hours after the release, respectively, while those on the left and right use NARR and ECMWF reanalysis fields, respectively. The color scale shows the logarithm of the SF$_6$ concentrations between $10^0$ ng m$^{-3}$ and $10^5$ ng m$^{-3}$.





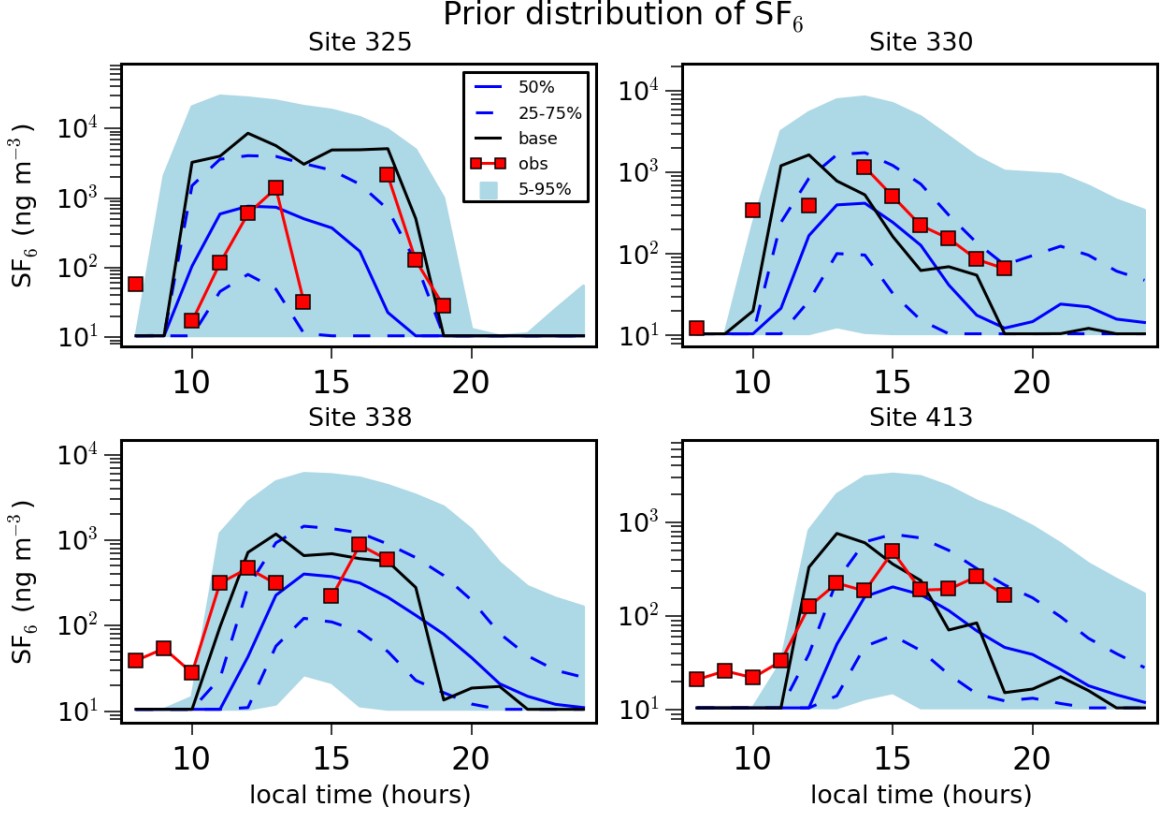

**Figure 7.** Time series of the prior probability distribution of $SF_6$ from the Latin hypercube ensemble of FLEXPART simulations at four representative measurement locations (see Fig. 3). Different quantiles of the probability distribution are displayed (blue lines and area), as are the base WRF-FLEXPART case using default and actual input values (black line), and the Diablo Canyon measurements (red squares). Local times correspond to the end of the one-hour average intervals.

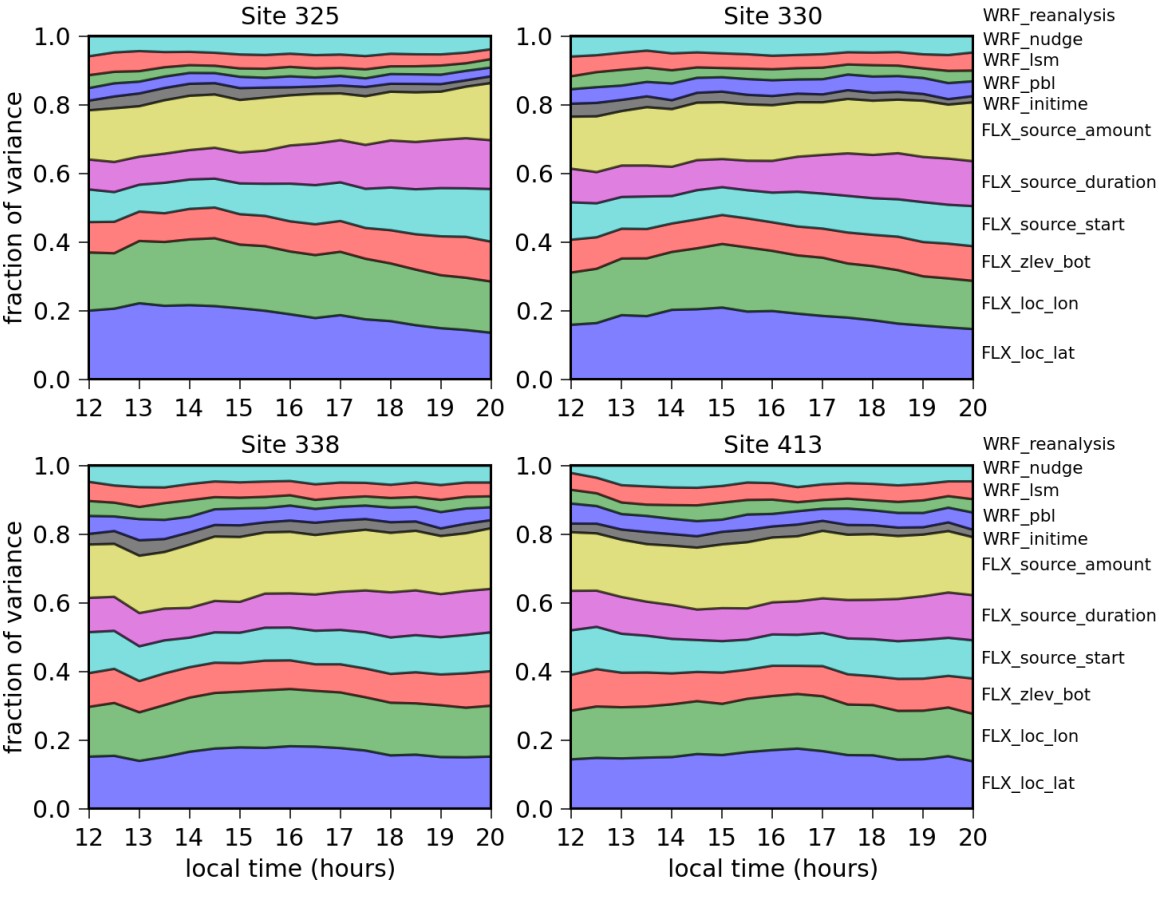

**Figure 8.** Time series of the SF$_6$ feature scores at the four representative measurement sites. Each colored band represents the fraction of the variance in the Latin hypercube ensemble caused by the parameters labeled on the right hand side. The WRF parameters are the upper five bands, while the FLEXPART parameters are the lower six bands.





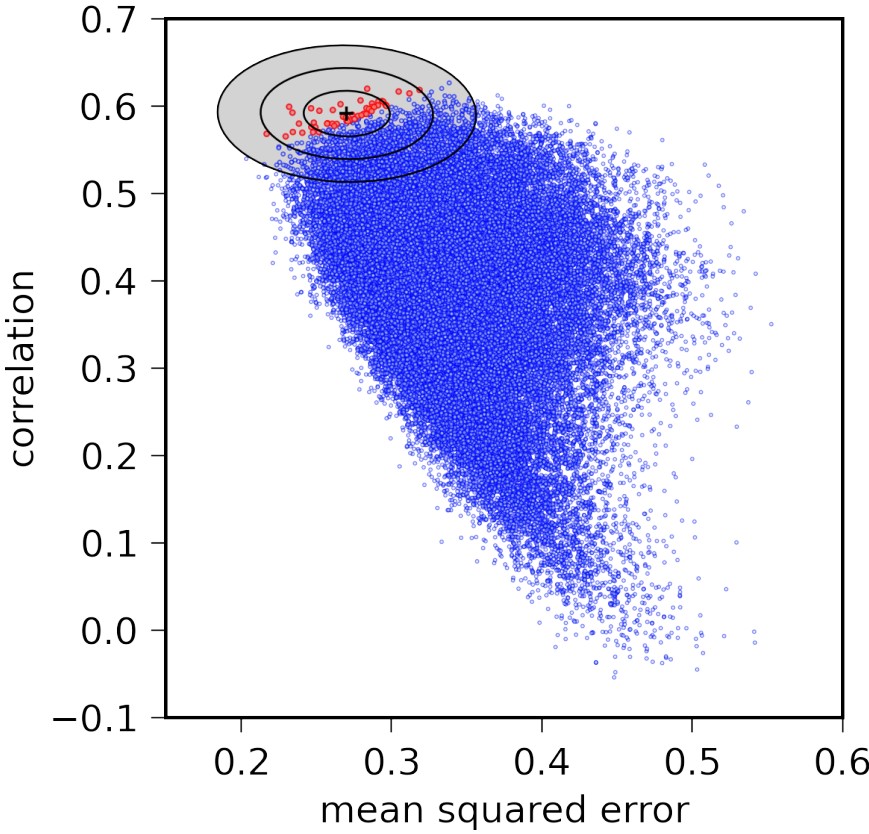

**Figure 9.** Correlation and mean-squared-error between the simulations and measurements of $SF_6$ for the 40,000 members of the WRF-FLEXPART ensemble (small blue dots). The upper left portion of the figure shows the 50 best fitting simulations (red dots), and the estimated target and covariance (1-to-3 standard deviation contours) in the likelihood function.





**Figure 10.** The marginal posterior distribution of WRF and FLEXPART parameters for the synthetic data inversion. Diagonal components show univariate continuous distributions for FLEXPART (top left) and univariate categorical distributions for WRF (bottom right). Off-diagonal components show bivariate distributions for the pair of parameters in the corresponding row and column. Probability density is normalized, with red colors denoting regions of high probability in the bivariate distributions. Known input values are shown by the black lines and circles in the diagonal and off-diagonal components, respectively.





**Figure 11.** The marginal posterior distribution of WRF and FLEXPART parameters for the Diablo Canyon tracer data inversion. Actual source term input values used in the tracer release experiment are denoted by the black lines and circles. Refer to the caption in Fig. 10 for further information.

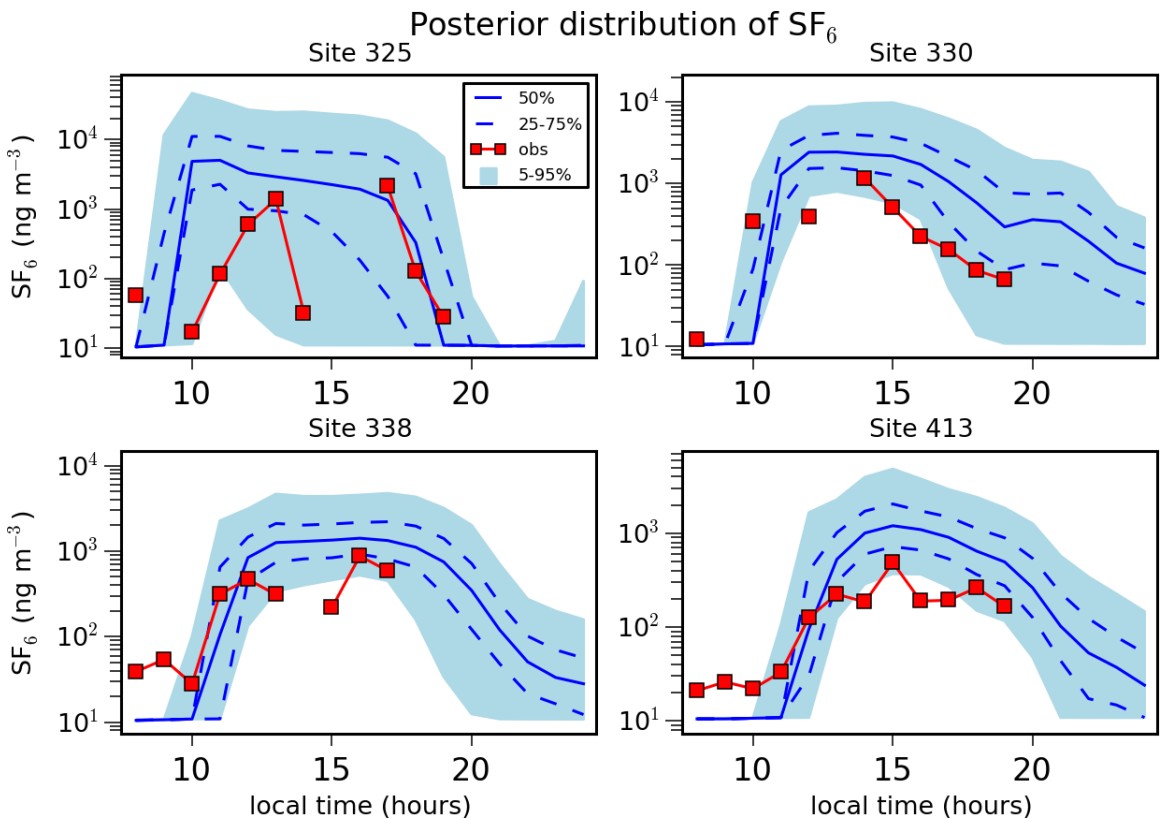

**Figure 12.** Time series of the posterior probability distribution of $SF_6$ at four representative measurement locations. Different quantiles of the probability distribution are displayed (blue lines and area), as are the tracer measurements (red squares).