# Peer review of "Bayesian inverse modeling of the atmospheric transport and emissions of a controlled tracer release from a nuclear power plant"

_Atmospheric Chemistry and Physics, 2017_

## Referee Comment (RC1) · Anonymous Referee #1 · 22 May 2017

**1   Overview:**

Review of "*Bayesian inverse modeling of the atmospheric transport and emissions of a controlled tracer release from a nuclear power plant*" by Lucas *et al.*

Lucas *et al.* present a comprehensive analysis of a novel inverse methodology for estimating the spatio-temporal location of a trace gas source. Their methodology is evaluated with a controlled release experiment. They are able to estimate the location of the controlled release with impressive spatio-temporal accuracy (their most likely results are within 200 meters of the known location, the release time is within 5 minutes, and the release duration is within 50 minutes). This is particularly impressive given

the complex topography of the region. This is an excellent manuscript. It is well written, uses state-of-the-art inversion techniques, and the figures are high quality. This manuscript should be published in ACP. The only comments I have are, seemingly, minor.

**2   Minor comments:**

**2.1   Using MSE and CORR?**

What is the advantage of using a bulk metric like MSE and CORR, why not use the measured concentrations? It seems that the problem is traditionally framed as the difference between modeled and measured tracer concentrations (the "model-data mismatch"). By moving away from that they instead have to specify a set of metrics to use. It seems to work quite well and I'm somewhat curious as to why.

**2.2   FDDA not useful in this case?**

It seems counter-intuitive that the FDDA would not be a preferred option in the inversion. This option *should* be incorporating more data and, as such, one would anticipate a better solution. Could the authors comment on this a bit more? For example, could this have to do with the complex terrain (maybe the FDDA doesn't help because it's only impacting some broader-scale flow)? Is using FDDA just adding computational expense that doesn't actually benefit us much at these fine spatial scales?

**2.3 Could do an inversion using real release parameters to provide information on WRF-FLEXPART modeling error?**

It seems that the authors could also use these same experiments to provide to useful information on the WRF-FLEXPART modeling error and the optimal setup. The authors could do an inversion where they use the known tracer release parameters and just invert for the WRF-FLEXPART setup. Would this give the same setup that was chosen in their current manuscript? Is FDDA still not the optimal setup?

This seems like it should be fairly straight forward to do with their setup and it could be quite useful for the broader community.

**3 Specific comments:**

Latin hypercube sampling: I'm not familiar with this sampling method, what was the motivation for using this one? Is it particularly well-suited to this problem?

Figure captions: Some of the figure captions could use a little more labeling. For example, Fig. 3 doesn't say what the coloring indicates (presumably it's showing elevation).

---

## Referee Comment (RC2) · Anonymous Referee #2 · 14 Jun 2017

This paper evaluates an inversion method using Regression Trees to estimate the strength and location of a tracer release. The method uses measurements from 125 sites around a release site during a field campaign and estimates both the uncertainty in the release parameters and the optimal WRF parameters for the meteorological simulation. The paper is thorough and well described and appropriate for ACP.

Major Comments:

I think it would be of interest to perform a synthetic run using the actual release parameters. In this way, the model could identify the uncertainty due to the simulations alone, which would be of great interest.

[Figure]

I would like to see a table for the synthetic test and the actual test, showing the true parameters and the model estimates. I know that this is already shown graphically in Figures 10 and 11, but a table of actual numbers would help.

I am concerned that for some parameters (esp. release height), the inversion range (Table 2) is too narrow. In Figs 10 & 11, some of the histograms continue up to the limit, and hence the results could be due to the input range rather than the inversion itself. I think that an expanded range should have been used – if this is still possible, it would be good to update the paper.

It would be very interesting to see the sensitivity to the model resolution. Maybe a future study could add as a parameter the number of domains used by WRF-FLEXPART. This would show how much is gained by performing a 300m resolution WRF simulation.

Were any sensitivity tests performed with the number of observations? There are few studies that have 125 measurement sites – this dataset could be used to identify how many stations are needed and what their optimal placement is.

Would it be possible to add a pair of scatter plots: one with the optimal parameters and one with an ordinary configuration (given that there is no a priori best guess that is different from the actual source parameters). How well does the model perform overall? How much improvement can be gained by adjusting WRF parameters and how much by adjusting the source parameters? This would be a supplementary angle on the question being addressed in Fig 8.

Minor Comments:

Abstract: I would recommend expanding the description of the WRF results in the abstract since determining the optimal set-up based on measurements is of great interest. In the abstract, I would also recommend providing some context for the accuracy of the estimate – number of stations and average distance. Without that, it is not possible to interpret the significance of the errors in the estimated source parameters.

Fig 1 caption: "used to improve the differences between simulations and observations." I think this could be rephrased. In fact, the whole caption could be improved.

I'm not sure that I could understand the discussion of the feature scores based on the explanation provided. A few more details would be welcome.

There are examples of boosted regression trees in atmospheric sciences, eg. Sayegh et al., and references therein. I think it would be good to relate the present study to some of those. There are also other studies using FLEXPART for source inversions. I would recommend adding a couple to the introduction and explaining the relationship to the present study.

Sayegh, A., Tate, J. E., & Ropkins, K. (2016). Understanding how roadside concentrations of NOx are influenced by the background levels, traffic density, and meteorological conditions using Boosted Regression Trees. Atmospheric Environment, 127, 163-175.

---

## Author Comment (AC1) · 14 Aug 2017

**General response to Referee #1**

We highly appreciate the feedback from Referee #1. The referee recommended a few minor technical corrections and commented on three issues. The issues are related to 1) the *mse* and *corr* metrics used to optimize model and measurement differences, 2) the apparent detrimental effects of nudging and data assimilation on the simulations, and 3) the information on modeling errors obtained by plugging in the known source values (location, time, and amount). We address these issues in further detail below

[Figure]

and will revise the manuscript for clarity and content based on the reviewer feedback. Again, we thank the referee for the informative comments.

**Minor comments**

1. *mse and corr metrics*
   The reviewer asks about the advantage of fitting the mean-squared-error and correlation metrics instead of the actual concentrations for the optimization. The advantage is mainly one of statistical convenience. We could, in principle, fit statistical models to the actual concentrations (e.g., using gradient boosting), and then use the statistical models to minimize concentration differences with observations. Fitting the concentrations is more challenging, however, because the statistical model becomes functional and the size of the problem is larger. Referring to Eqs. 7 and 8 in the manuscript, our vectors and covariance matrix are only two dimensional (*mse* and *corr*). The terms in these equations would have more dimensions if we directly modeled the concentrations (1,148 without applying a dimensional reduction technique like principle components analysis). Further, we would have to estimate spatial and temporal correlations in the covariance matrix. We have collaborated with statisticians to tackle this more challenging statistical problem (Francom et al., 2016). This effort has shown that the additional complexity of modeling the concentrations can pay off in the form of tighter parameter estimates.

2. *Nudging effects*
   Like the reviewer, we find it counterintuitive that increased levels of meteorological nudging appear to degrade the agreement with concentration observations. Given the relatively small likelihood differences between the nudging options though, this result is not highly robust and not one of our major conclusions. Nonetheless, we also surmise that tradeoffs between nudging, variability, and

physics parameterizations in complex terrain may be playing a role. At higher levels of nudging, there is less variability, and therefore less opportunity for parameterizations to compensate for terrain-related errors. At low levels of nudging, there is more variability and a greater possibility to find better matches with observations. In future work it would be interesting to further diagnose nudging in complex terrain.

3. *Model error from known source*

The reviewer recommends adding results using the known source parameters to help quantify model error. Both reviewers made the same excellent suggestion, so we expanded Sect. 5.1 in the revised manuscript to accommodate these results. A new scatterplot in the section shows the mean squared error and correlation for the 162 WRF runs with the known source parameters. These variations are not as large as they are across the full 40,000 member ensemble, though they are still significant. We also show that the known source variations are explained mainly by differences in the reanalysis data sets and initialization times, which agrees with our previous variance analysis in Sect. 5.3. Other factors, including nudging levels, do not play a very large role.

**Specific comments**

In addition to the above comments, we will also revise the manuscript by

- providing a better describing of Latin hypercube sampling,

- and improving figure captions (e.g. Fig. 3).

---

## Author Comment (AC2) · 14 Aug 2017

**General response to Anonymous Referee #2**

Anonymous referee #2 provided valuable feedback and comments about our ensemble analysis and inversion. The referee had six major comments, four minor comments, and recommended citing other studies using boosted regression trees (e.g. Sayegh et al). We greatly appreciate the reviewer's thoughtful comments, especially the suggestions to include additional information and analysis about the WRF variability using known source parameters. We address each of these comments below and

have revised the manuscript accordingly. The revised manuscript includes descriptions and analysis for three new figures and modifications to an existing figure and table. We believe that these revisions are sufficient to address the reviewer's comments for publication in ACP.

**Replies to Major Comments**

1. *Analysis using actual release parameters*
   The reviewer recommended additional analysis using the known source parameters to highlight the variability due to WRF alone. The submitted version did not include this analysis because we wanted to show that the source inversion algorithm works well without knowledge of the source. That is, we excluded known source simulations from the training data as a rigorous test of our method. However the reviewer's recommendation is important and valuable, so we have expanded Sect. 5.1 in the revised manuscript to describe the known source analysis. This section includes two new figures showing the variability due only to WRF when the source parameters are known. One of the new figures, which is similar to Fig. 7 in the submitted version, shows the ensemble spread caused by the differences among the 162 WRF runs in the time series of $SF_6$ at the four selected locations. The other new figure shows the mean squared error and correlation for the 162 WRF runs using the known source parameters. Fig. 9 in the submitted version has also been modified to show the WRF-only variations relative to the larger Latin hypercube ensemble variations. The results of the new analysis are consistent with the findings described in the original manuscript. The WRF-only variations are important, though not as dramatic as the full ensemble variations. Further, the categorical variables in WRF related to reanalysis data and initialization time explain most of the spread within the known source WRF-only ensemble.

2. *Table of actual and estimated parameters*
   The reviewer requested a table that compares the source parameters estimated by the inversion algorithm to the actual values used for both the synthetic and real tracer releases. Table 3 in the submitted manuscript already shows the comparison for the real tracer release. This table is small and cross-referenced toward the end of the conclusions section, so it is not surprising that the reviewer missed it. In the revised manuscript, we expanded the table by adding the comparison for the synthetic tracer release and cross-reference it earlier in the discussion (in Sect. 5.5 and 5.6).

3. *Prior sampling ranges & insensitivity to release height*
   The reviewer notes that some of the sampling ranges for the source parameters in the prior distribution, especially the release height, seem to be too narrow because the posterior histograms extend to the limits. This behavior is not necessarily undesirable because the source parameters can be estimated from likelihood maxima that occur within the limits. Moreover the widths of the posterior distributions provide information about the uncertainty in estimated source parameters. Wide posterior distributions can result from observations that are not constraining or a model that is insensitive to parameter variations.

   For the release height, we used a narrow range to represent near-surface releases. We are not surprised that the simulations are not sensitive to the narrow variations because the reanalysis data sets and model simulations use a relatively coarse vertical resolution. We therefore expected to see similarities between the posterior and prior distributions for the release height (i.e., both are "flat"), which was confirmed by our results. It is often useful to include non-sensitive parameters in this type of Bayesian analysis to verify that the algorithm works properly.

   As the reviewer suggests, it would be interesting to broaden the sampling range for the release height to also cover elevated releases. However, this would re-

   quire running a large number of new ensemble simulations, which we are unable to perform at this time (note that narrowing the range would not require new simulations). We will revise the manuscript to better describe the relationship between the posterior widths and parameter sensitivities, with an emphasis on the release height. We will also mention our interest to add elevated releases in future work.

4. *WRF resolution*
   The 300-meter resolution of the innermost WRF domain was selected to resolve some of the terrain features surrounding Diablo Canyon. For instance, a line of measurement sensors detected $SF_6$ in a small canyon stretching from location 413 to the northwest (see Fig. 3). The selected model resolution is a compromise between being able to simulate the transport of $SF_6$ through these features and having scale-dependent parameterizations that are still valid.

   Not all of the sensor locations require 300-m resolution, however, so the reviewer's suggestion to incorporate WRF resolution into the analysis is very interesting and relevant. We could potentially speed up the inversion by initializing with coarser WRF resolutions and progressing to finer resolutions. We could also quantify the joint effects of resolution differences and parameter variations. Though we haven't conducted this analysis for WRF and FLEXPART, we have done something similar for a global climate model. In that work, we ran ensembles at different resolutions and included resolution as an input parameter in our machine learning models. We will include additional discussion about WRF resolution in the revised manuscript. We also welcome future collaborations with the reviewer and others on this interesting topic.

5. *Number of observations*
   The Diablo Canyon field study was heavily instrumented because of the complexity of the surrounding terrain and due to close proximity to populated areas. We agree that this data would be valuable for optimizing the effects of measurement density and location. Although we have conducted optimal analysis of this type

for other studies and regions (e.g., see Lucas et al., 2015), we did not perform a similar optimization here. Instead, we roughly assessed the impacts of reduced data density by selecting fractions of measurement points at random, for fractions between 5% and 90%. We then re-computed the $R^2$ coefficients of determination for the gradient boosting regression models, as described in Sect. 5.4. As fewer measurement points are used in the fits, the quality of the GB models degrade, which, in turn, limits the ability to perform an inversion. From this analysis, using only 20% of the measurement points still results in $R^2$ values of about 0.82 and 0.96 for the *mse* and *corr*, respectively, which should be sufficient for estimating the most influential source parameters. Because some measurement locations are more important than others, we should be able to reduce the fraction of measurements below 20% without adversely affecting the inversion, if we optimize the locations. We will include a summary of this discussion in the revised manuscript and consider future observational optimization studies for Diablo Canyon.

6. *Optimized versus regular parameters*
   We agree with the reviewer that it is a good idea to include a new figure that compares good and bad model simulations to the measurements. We selected high and low likelihood cases from the 40,000 member Latin hypercube ensemble and compare their $SF_6$ concentrations to measurements. This figure clearly shows that the high likelihood case provides a much better match to the measurements than the low likelihood case. It also shows that simulations have a positive bias, which leads to the minimum *mse* of about 0.2 instead of 0. This figure will be included and described in Sect. 5.4. Along with the modified version of Fig. 9 described in item 1 above, these figures will show the improvements that can be gained by varying WRF parameters and source parameters.
* * *
**Replies to Minor Comments**

As suggested by the reviewer, the revised manuscript will provide

- more details in the abstract about the optimized WRF configuration and accuracy of the parameter estimates,

- an improved caption for Fig. 1,

- further discussion about feature scores and parameter sensitivity,

- and additional references and information about gradient boosting applications in atmospheric science.